# How Do Different Modes of Governance Support Ecosystem Services/Disservices in Small-Scale Urban Green Infrastructure? A Systematic Review

**Sina Razzaghi Asl** [1,*] **and Hamil Pearsall** [1]

Department of Geography and Urban Studies, Temple University, Philadelphia, PA 19122, USA
* Correspondence: tuk98809@temple.edu

**Abstract:** As cities are facing environmental and societal challenges, including climate change, rapid urbanization, and the COVID-19 pandemic, scholars and policymakers have recognized the potential of small-scale urban green infrastructures (UGI), such as rain gardens and street trees, to support important ecosystem services (ES) during periods of crisis and change. While there has been considerable research on the design, planning, engineering, and ecology of small-scale UGI, the governance modes of such spaces to support ES and manage ecosystem disservices (EDS) have received significantly less research attention. In this article, we provide a systematic review to evaluate how different modes of governance support different ES in small-scale green infrastructure. We evaluated governance in six types of small-scale green infrastructure: small parks, community gardens, vacant lands, rain gardens, green roofs, and street trees. Our review examines the different characteristics of four new governance approaches, including adaptive, network, mosaic, and transformative to understand their bottom-up nature and applicability in governing ES/disservices of small-scale UGI. Each governance mode can be effective for managing the ES of certain small-scale UGI, given their associations with principles such as resilience thinking, connectivity, and active citizenship. Our synthesis highlights knowledge gaps at the intersection between governance arrangements and ES in small-scale UGI. We conclude with a call for further research on the environmental and contextual factors that moderate the linkages between governance modes and ES/EDS in different types of UGI.

**Keywords:** governance; small-scale UGI; ES; ecosystem disservices; UGI principles; systematic review





## 1. Introduction

Urban land area is predicted to triple from 2000 to 2030 [1]. This rapid urbanization will negatively impact local and regional ecosystem functions and will exacerbate consequences of the climate change, and reduce the adaptive capacity of urban areas to cope with a changing climate [2,3]. Urban green infrastructure (UGI) such as green roofs, green facades, public parks, urban forests, urban wetlands, and unmanaged green sites [4], provide nature-based solutions (NBS) that offer a promising avenue for climate change adaptation in cities to reduce the negative environmental impacts of urbanization, such as the urban heat island effect and altered precipitation patterns. UGI supports a wide range of ES at different spatial levels including but not limited to provisioning (e.g., food, and freshwater), regulating (e.g., urban temperature regulations, noise reduction, air purification, pollination, runoff mitigation, and waste treatment), socio-cultural (tourism, recreation, cognitive development, social cohesion), and supporting (e.g., habitat for biodiversity diversity), with fewer documented health benefits (e.g., good health, mortality) [5–13]. Research has also documented EDS associated with UGI, including conflicts with grey infrastructure, air pollution, and green gentrification [14–16].

The management and governance of UGI provide an important mechanism for balancing the services and disservices of UGI in cities. Governance can be defined as a process of collective decision making that allows different stakeholders to include their needs and

expectations [17]. Different governance approaches have been widely used in the field of natural resource management to protect and sustain resources [18–20]. There has been an increase in scholarship on UGI governance with growing recognition of the importance of UGI for addressing stormwater runoff, urban heating, and air pollution. Governance is a critical component of effective UGI implementation as cities experience significant changes such as extreme events, pandemics, and biodiversity loss [21]. The role of UGI may also undergo abrupt, surprising change [22,23]. For example, in response to government restrictions such as stay-at-home, social distancing, and quarantine policies during the recent COVID-19 pandemic, UGI became an important resource for socio-cultural and regulatory ES because green spaces provided physical and mental health benefits [24]. However, given urbanization trends, as well as environmental change in existing urban areas, there is a need for cities to develop suitable environmental governance approaches actively and intentionally to address pressing societal challenges [25]

We review governance approaches in the context of small-scale UGI to evaluate how different governance models address ES. UGI governance is defined here as the "processes, interactions, organizations, and decisions" related to greenspace provision and administration, as defined by Lawrence et al. [26]. Although there is no consensus to define UGI, one of the most recently published studies indicates UGI as a multiscalar concept, which can range from small-scale green infrastructure such as rain gardens, pocket parks, community gardens, and green lands, to large-scale facilities targeting the protection and preservation of the natural habitats [27]. Small-scale UGI offers a unique opportunity to enhance ES while minimizing disservices [28–31]. First, the effectiveness of UGI is largely dependent on interconnected social and ecological processes that need to be properly managed and planned at the local scale while also connected to broader scale policies. Because small-scale UGI is often decentralized and has very different governance processes from large-scale UGI [32], the management and governance of small-scale UGI can be more responsive to local social and ecological needs. Second, considering that large- and medium-scale green areas are usually covered with a large area of single-species allergenic species in cities, highly diverse small-scale green spaces can significantly reduce the risk of allergenicity from urban green spaces. [33–35]. Furthermore, several recent studies have suggested that large- and medium-scale green spaces are associated with gentrification outcomes, whereas small-scale green spaces may limit increases in property values [36–38]. Finally, despite a wide variety of literature on the design, planning, engineering, and ecology of small-scale UGI, the governance dimensions of such spaces to support ES have, to date, received significantly less research attention [39,40].

Over the last few decades, many countries have developed their governance practices to optimize ES in small-scale UGI to cope with growing challenges such as water scarcity, biodiversity loss, institutional shortcomings, citizen participation, fiscal austerity, shortcomings of top-down management, lack of environmental knowledge, lack of political stability, and mismatch between boundaries and the scale of ES [40–43]. However, there is little research that compares how different governance approaches address ES in small-scale green infrastructure. To address this gap, we synthesize literature on small-scale UGI management and governance, drawing on diverse geographic contexts to provide a better understanding of existing approaches' characteristics and exploring the conditions under which UGI governance approaches may emerge. We do not aim to investigate the suitability or adaptability of these approaches with different UGI types. Instead, we provide a synthesis of how different governance approaches address ES in the six studied types of small-scale UGI.

## 2. Literature Review

### 2.1. Urban Green Infrastructure and ES

Green infrastructure (GI) is a relatively new concept, and several studies have proposed different definitions for GI. The two most cited definitions are from Benedict [44] who defines GI as "an interconnected network of green space that conserves natural ecosys-

tem values and functions and provides associated benefits to human populations", and the European Commission [45], which defines GI as "a strategically planned network of natural and semi-natural areas with other environmental features designed and managed to deliver a wide range of ES. It incorporates green spaces (or blue if aquatic ecosystems are concerned) and other physical features in terrestrial (including coastal) and marine areas. On land, GI is present in rural and urban settings". According to these definitions, key characteristics of GI, including multifunctionality, ES, ecological networks, connectivity, and multiscalar, serve as boundary concepts among various policymakers, planners, and researchers to guide UGI planning and designing [46].

Urban ecosystem services (UES) have multiple benefits for human health and well-being in the face of rapid urbanization, land-use transformation, and climate change crisis [47]. ES can be defined as "the benefits people obtain from ecosystems" [48]. UES is supported by a diverse green infrastructure type including but not limited to parks, urban forests, farmlands, vacant lots, and gardens. UES can be divided into four categories according to the Millennium ecosystem assessment [48]: provisioning services (materials obtained from ecosystems), regulating services (benefits obtained from the regulation by ecosystem process), habitat or supporting services (essentials to produce all ES) and cultural services (non-material benefits obtained from ecosystems). Research suggests that small-scale green infrastructure can moderate the negative environmental impacts of rapid urbanization and climate change by contributing to recreation, mitigating air pollution, cooling surface, and air temperatures, and retaining stormwater run-off [49]. For example, green roofs and walls may improve air quality and flood control management or street trees can reduce exposure to pollution in urban areas [14,50,51]. Moreover, community gardens in urban neighborhoods not only provide food but can also have health, social and aesthetic benefits for the local community [52–54]. Green spaces and urban trees can also mitigate air temperature through transpiration, evaporation, shading, and modifying wind-flow mechanisms [55]. A study by Peschardt et al. [56] indicates that small-scale green spaces have socializing benefits because they provide spaces for neighbors to interact, whereas other services such as noise reduction and carbon storage are less associated with small-scale green spaces compared with large-scale green infrastructure due to their lower compactness or density.

Table 1 summarizes the ES provided by six types of small-scale GI examined in this study. As can be seen, small-scale UGI provides a wide variety of benefits, albeit some UGI types, such as community gardens, may provide a larger range of services than others. A review of articles, in this case, shows that most studies have focused on green spaces' benefits and the impacts of some types of UGI, such as rain gardens or pocket parks on human health, microclimate regulation, and socio-cultural services. In contrast, few studies have examined how different modes of governance might shape ecosystem service provision in different types of UGI.

**Table 1.** Urban ES provided by six studied small-scale green infrastructures.

| ES | Some Examples of UGI and Their Impacts in Literature |
|---|---|
| Provisioning | Community gardens can address food security in urban areas [57,58] |
| Supporting | Street trees offer key conservation opportunities for pollinators [59], they also reduce the negative effects of urbanization on birds [60]; green roofs can have ecological significance by attracting and supporting urban fauna [61]; vacant lands can support insects' habitats [62] |
| Regulating | Vacant lands have cooling effects in urbanized areas [63]; green roofs have large impact on the urban heat island effect, positive effect on street canyon air quality, and stormwater management [64–66]; rain gardens may provide considerable carbon potential, offsetting the whole carbon footprint [67]; street trees can reduce air quality depending on the aspect ratio as well as stormwater [68,69]; community gardens can reduce surface runoff [70] |

**Table 1.** *Cont.*

| ES | Some Examples of UGI and Their Impacts in Literature |
|---|---|
| Socio-Cultural | Small parks offer health benefits [62,71]; green roofs offer recreational and experimental benefits for residents [72]; community gardens as learning environments for sustainability [73]; vacant lots may provide social and cultural values for local communities [74] |

### 2.2. Urban Green Infrastructure and EDS

While UGI has several benefits, it also sometimes produces EDS that are frequently overlooked [75]. The concept of EDS refers to the negative impacts that ecosystems can have on humans and their environs [76]. According to Lyytimaki and Sipila [76], EDS are "functions of ecosystems that are perceived as negative for human well-being" and can be brought on by natural or political occurrences such as floods, earthquakes, wildfires, or conflicts. For example, small-scale UGI such as street trees may provide allergies associated with grass pollen and damage to properties [77–80]. Some species release a significant amount of biogenic volatile organic compounds (VOCs), which, when combined with nitrogen oxides (NOx), can create particulate matter, secondary organic aerosol, and ozone, which exacerbate respiratory diseases such as asthma [81]. In addition, research shows that the risk of vegetables and soil contaminated by heavy metals and pollutants in community gardens and green roofs can be considered EDS [82]. There is no agreement on how to classify EDS in relation to ES, despite the fact that certain research has split it into various groups [73,83–86]. Better understanding of the conditions under which EDS arises will help policymakers, practitioners, and communities reduce these negative impacts. While urban areas depend on ES, understanding disservices are of paramount importance from a governance lens. Since EDS reduces public support for UGI, it is important to reduce these negative impacts to optimize UGI for sustainability. For example, in the Mediterranean region, the ornamental patterns of the urban areas imply significant pollen risk from woody species such as plane trees or cypresses, as the most allergenic ornamental species [87,88]. Some studies such as those conducted by Von Döhren and Haase [72] and Sousa-Silva et al. [88] have provided a reliable overview of the environmental and health issues produced by different types of urban trees. Table 2 summarizes some examples of EDS provided by six types of small-scale GI examined in this study.

**Table 2.** Urban ecosystem disservices are provided by six studied small-scale green infrastructures.

| EDS | Some Examples of UGI and Their Impacts in Literature |
|---|---|
| EDS | Tall and leafy trees may block the views [89];<br>Vacant lands may be unsafe and ugly [89];<br>Some plant species may create allergenic pollen [90–92];<br>Tree roots may cause sidewalk pavement problems [93];<br>Community gardens may get contaminated by greywater irrigation from contaminated drainage channels or streams [94];<br>Increasing UGI results in an increase in hornet species [95];<br>Urban trees produce green waste resulting in public health issues [14] |

### 2.3. A Need for New Governance Approaches

Enhancing urban resilience and sustainability in the face of "wicked problems" are key challenges for UGI governance [95]. According to Andersson et al. [96] and Jerome [97], small-scale UGI can contribute multiple co-benefits to support a wide variety of ES. However, there are still some barriers and uncertainties to governing and managing different types of GI worldwide.

One of the important challenges for governance in existing small-scale UGI, such as pocket parks or vacant land uses, is that they can be temporary or short-term land uses. For example, a study in Detroit, Texas found that ragweed populations are more

common in vegetated vacant lots, making the transition management of these lots crucial to avoiding significant effects on allergenic pollen burdens [98]. Thus, if cities rely on the ES that these spaces provide, there is a need for governance mechanisms that either provide long-term security for these spaces or support a more adaptive, flexible, and dynamic governance approach to cope with the temporary negative consequences of these spaces [99,100]. Kabisch [101] states the major challenges for green infrastructure governance in Berlin as financial constraints, loss of expertise, and low awareness of such spaces' benefits at the local scale. Fox-Kamper et al. [102] found the major barriers to community garden governance include unsecured land tenure, community engagement, and lack of long-term governance support. A study by Guitart et al. [103] shows that the main challenge for community garden governance in the United States is land tenure where gardeners lack long-term access to land. Furthermore, some scholars have highlighted the issue of changing governance settings and GI data inconsistency as some of the most important challenges GI are facing [104]. Undoubtedly, one of the most important barriers to implementing GI, such as rain gardens, is their costs. These facilities can be expensive to install and maintain, which in turn reduces the willingness of planners and owners to shift toward them [25].

Moreover, urban governance is challenging given environmental justice (EJ) issues in terms of UGI equitable distribution, transparent procedures, and sufficient recognition of various actors' needs and perceptions [2]. Availability, accessibility, and attractiveness of small-scale UGI for different social groups and inhabitants are among the most important issues that EJ research has recently addressed [105,106]. For instance, Sanchez and Reames [107] address spatial equity in green roof distribution in Detroit, MI, and show that green roofs were concentrated in the wealthiest part of Detroit's urban core with a predominantly white population. Consequently, an emerging focus in environmental governance is how different governance approaches can broaden access and participation to diverse social groups, particularly marginalized or vulnerable groups. A potential opportunity for small-scale UGI to promote environmental justice lies in its need for local governance, which can place decision making in the hands of local communities and give them ownership over these spaces. In addition to promoting equitable governance, local ownership may reduce disservices, such as green gentrification, which has been identified as a concern by researchers and non-profit sectors in recent years [108]. In other words, an equitable distribution, experience, and understanding of UGI throughout the cities is an important goal of UGI governance.

Significant shifts have occurred within environmental decision making on UGI in the past 20 years. These shifts have yielded collaborative and bottom-up management approaches to guarantee future success in the face of rapid urbanization, climate change, and major societal disruptions, such as the COVID-19 pandemic. UGI government styles based on centralized decision making, public budgets, top-down, and bureaucratic arrangements have been replaced increasingly by new horizontal approaches of environmental decentralized governance focused on the fluidity between top-down and bottom-up approaches. This new emerging paradigm shift largely emphasizes the concepts of flexibility, collaboration, coordination, awareness, adaptation, inclusiveness, knowledge generation, and transparency [109–113]. As a result, a range of new democratic governance approaches is in use under conditions of uncertainty, complexity, instability, and unpredictability to include different stakeholders' voices in the UGI decision making process and problem-solving. The uncertainty and complexity of managing ES at the local scale is related to socio-political (e.g., population growth), economic pressure (e.g., shrinking budgets), and environmental changes (e.g., climate change). New UGI governance approaches are intended to better address multiple stressors of urbanization and climate change by utilizing ES and harnessing disservices [19,38,114–116]. Over the last several decades, a wide variety of governance arrangements have been proposed, including "state governance" of publicly owned vacant lands and community gardens, and "networked governance" of public-private partnerships for local parks to the "self-governance/market-based" approach of

guerilla gardening. However, examining the applicability of different new governance approaches and policies to co-create and co-manage UGI is an important research direction.

## 3. Research Methods

### 3.1. Study Selection

This study conducted a systematic literature review on new governance approaches of small-scale UGI following the PRISMA procedure introduced by Moher et al. [117]. We conducted a systematic review to address the need for a review, critique, and possible reconceptualization of the diverse and interdisciplinary knowledge base on UGI, ES, and governance approaches [118]. The methodological approach used PRISMA key processes to construct the sampling frame, as shown in Figure 1: study planning and identification, screening and selection of publications, and content analysis of the selected documents.

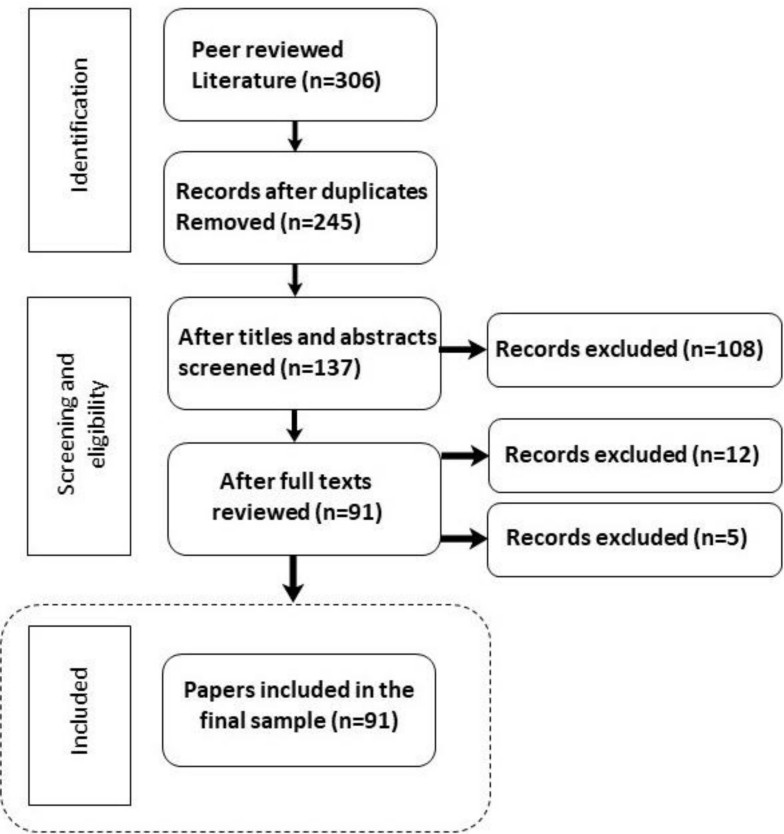

**Figure 1.** Flow diagram showing the methodology based on PRISMA procedure adapted from Moher et al. (2009).

First, a set of keywords in Google Scholar were used to identify studies on small-scale UGI and governance in urban settings. Six different small-scale UGI were selected, including small parks, community gardens, vacant lands, rain gardens, green roofs, and street trees in this article (Figure 2). All different combinations of six UGI were searched using Scopus and the Web of Science (WOS) databases as the search engine, with the search field set to 'keywords', and the document type set to 'article' or 'review'. To find relevant literature, five separate search queries (garden included both rain garden and community garden) were used, each with a different two-way combination of (keyword category-related) search phrases (Query 1: Park AND governance; Query 2: Garden AND governance; Query 3: Vacant land/lot AND governance; Query 4: Tree AND governance; Query 5: Green roof AND governance). To consolidate and deepen the review, the second round of searches was begun by setting the search field to a two-way combination of (green space/governance) and (nature-based solution/governance). The year of the publications was not filtered; hence, the sample collected from the databases contained all years of publication.

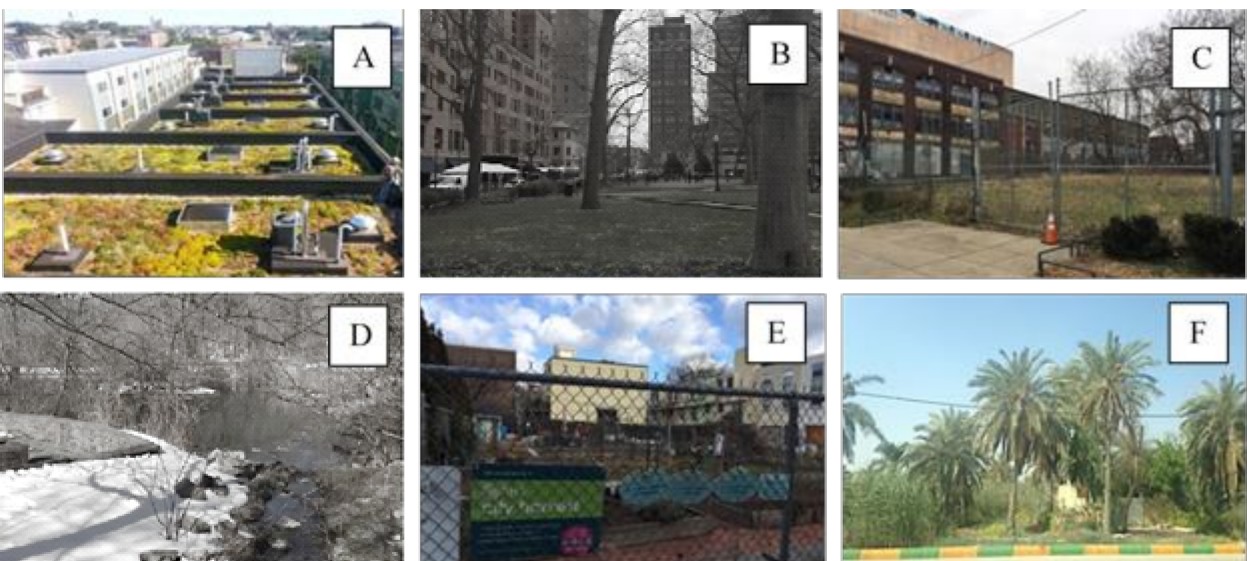

**Figure 2.** Diverse small-scale urban green infrastructure. (**A**) A small green roof on top of a residential building, USA. (**B**) A local public park, USA. (**C**) A privately owned piece of vacant land, USA. (**D**) Rain garden development, USA. (**E**) A community garden, USA. (**F**) Street trees, Iran. Photos taken by the authors.

*3.2. Literature Synthesis*

The search returned a total of 306 articles. A selection of 245 articles was found after duplicates were eliminated and they were ready for analysis. Since this review seeks to investigate only bottom-up governance approaches, traditional top-down, government-led arrangements were excluded from the final sample. In total, 108 articles were disqualified in the first round of screening (at the level of the title and abstract). The resulting sample's titles, abstracts, and full text were assessed for relevancy considering only six types of small-scale UGI and new governance modes. Since this study addressed the link between ES and UGI, only articles focused on different types of ES through governance approaches were included. Articles with a rural or regional focus and large green infrastructures such as urban parks or vast vacant lands were excluded. Two runs of full-text screening were conducted; the first resulted in the exclusion of 12, and the second pass resulted in the exclusion of an additional 5.

The systematic review was completed by analyzing 91 academic articles through qualitative content analysis [119]. Content analysis is a research approach for testing theoretical concerns and improving data comprehension in which a condensed number of concepts or categories characterizing a reality, a theory, or a study topic can be obtained [120]. Our review included publications with a focus on concepts and models for governance of ES in different types of small-scale green infrastructure. Since the objective of content analysis of sample documents is to highlight the applicability of new governance modes in the context of six small-scale UGI, the following questions guided the analysis: (1) What types of ES are associated with small-scale GSI? (2) What are the principles of new governance approaches associated with six small-scale UGI? (3) Which new governance modes were associated with different types of small-scale UGI? The authors were then able to determine the most prevalent governance modes and their core principles in the literature by carefully interpreting the articles and all the ensuing categories.

## 4. Results and Discussion

Based on our review of 91 articles, it was possible to identify four new, bottom-up governance approaches to small-scale UGI: adaptive, mosaic, network, and transformative. Despite some overlaps among these modes of governance, their nature and principles vary to some degree based on contextual, environmental, and social parameters. However,

the main similarity among these modes of governance was their bottom-up and multi-agent characteristics, which help policy makers find the most suitable solutions to manage complex urban green areas. The results of this review demonstrate that although the ES and governance modes are not directly linked, their potential relationships may be discerned through the identification of several principles that are applicable in different UGI cases. Some studies have directly focused on these principles, whereas others indirectly described them. Table 2 shows a set of principles for each governance mode which were applied regarding ES in studied small-scale UGI.

### 4.1. Date and Type of Studies

A systematic review of 91 articles indicated that the governance aspect of UGI is a growing topic of interest in academic research. The earliest articles were published in 2000 in the *Journal of Health and Places* about community gardening development and management. Based on their methods, the articles were divided into four groups: experimental, observational, discussion, and review (Figure 3). Discussion studies were primarily theoretical without data collection, whereas experimental studies involved intervention and primary data collection, observational studies used secondary data to analyze a phenomenon without intervention, and review studies conducted a systematic or scoping literature review. The most common study type was experimental ($n = 36$), followed by discussion ($n = 27$), observational ($n = 23$), and review ($n = 5$). In general, surveys, including interviews and questionnaires from stakeholders, were the most prevalent kind of experimental study. In addition, we found that community gardens and street trees were the most common types of UGI discussed in the reviewed articles. The specific characteristics of community gardens including but not limited to a socially inclusive, location in central parts of the neighborhoods, and a need to make fluidity between up-down and down-top management can lead them to be one of the most debatable themes in the UGI governance literature. Street trees were also amongst those UGIs that were largely reviewed thanks to their public nature, largely known benefits, and probably the fact that they are often apparently co-managed by both residents and municipalities. However, the adaptability and transformability of the governance modes of these elements in the face of societal and environmental changes (e.g., recent pandemic) can be further analyzed in different contexts to help policymakers to plan more mental and physical resilience for the future [119,120].

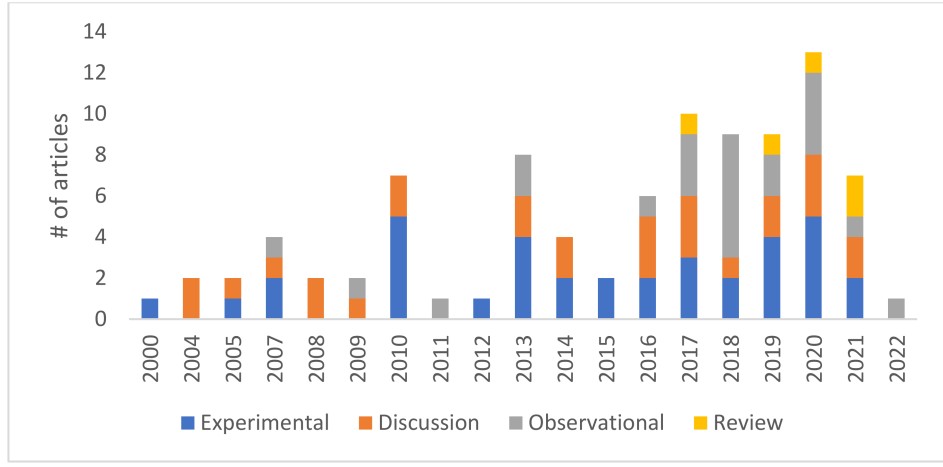

**Figure 3.** Publication dates and types for all articles ($n = 91$).

### 4.2. Governance Modes

Adaptive and co-management governance approaches were most reported in literature ($n = 48$), followed by other governance arrangements including mosaic ($n = 30$), network ($n = 10$), and transformative ($n = 3$). The three common principles used in literature belonged to adaptive governance including adaptability with the highest number ($n = 45$

of 91 adaptive groups), diversity of stakeholders (*n* = 39 of 91 articles), and flexibility (*n* = 36 of 91 articles), followed by self-governance as a core principle of mosaic governance (*n* = 29 of 91 articles). Moreover, the systematic review showed that the two principles of connectivity and diversity of stakeholders/actors emerged in two governance categories of adaptive/mosaic and adaptive/networked, respectively, making them more important in the case of decentralized and flexible UGI management.

The four governance modes and their principles are illustrated in Table 3.

**Table 3.** Four major governance approaches for six small-scale UGS and their principles in the literature.

| Governance Models | Principles | Number of Studies | |
|---|---|---|---|
| **Adaptive governance** | **Adaptability** is the capacity of actors to influence resilience. | 45 | 48 |
| | **Diversity of stakeholders** facilitates collaboration among institutions and jurisdictions. | 39 | |
| | **Flexibility** allows stakeholders to adapt their needs and expectations to new opportunities. | 36 | |
| | **Social learning** allows actors to share their values, experiences, and actions. | 23 | |
| | **Connectivity** facilitates negotiations and collaborations across horizontal (collaborative) and vertical (hierarchical) connections. | 22 | |
| | **Resilience thinking** is about how to learn to live with change and make use of it. | 17 | |
| **Mosaic governance** | **Self-governance** strengthens the autonomy of citizens to shape their own bottom-up initiatives and rules. | 29 | 30 |
| | **Active citizen** enhances the ability of people to organize themselves in a multiform manner. | 26 | |
| | **Polycentricity** allows multiple centers of governance to interact with each other across diverse scales and actors. | 26 | |
| | **Connectivity** fosters social and ecological resilience through linking actors. | 18 | |
| | **Stewardship** focuses on collaborative management activity. | 14 | |
| | **Reflexivity** allows to include the perspectives, values, and norms of a variety of actors. | 11 | |
| **Networked governance** | **Knowledge sharing** allows exchanging information between local stakeholders. | 7 | 10 |
| | **Social networks** facilitate social interactions between actors. | 7 | |
| | **Diversity of actors** allows the presence of various actors, often multi-level. | 6 | |
| | **Decentralization** transfers organization activities to several local actors. | 4 | |
| **Transformative governance** | **Social innovation** is the design of new solutions to imply transformative changes. | 3 | 3 |
| | **Transition management** accelerates the sustainability transition through the participatory process of visioning, learning, and experimenting. | 3 | |
| | **Regime shifts** are large, abrupt, persistent changes in the structure and function of ecosystems. | 2 | |
| | **Long-termism** allows improving the long-term future of ecosystems. | 2 | |
| | **Panarchy** means drastic transformative changes. | 1 | |

*4.3. Adaptive Governance*

Adaptive governance (AG) as the most frequent mode mentioned in the literature, is defined as "*an outgrowth of the theoretical search for modes of managing uncertainty and complexity in socio-ecological systems*" [25]. AG appears in almost more than half of the papers examined (48). Allen and Gunderson [121] define it as "*the institutional framework that deals with social and political dimensions of resource management and that allow adaptive management to function*". AG arrangements include a network of multiple public and private actors to cope with uncertainty and complexity in small-scale UGI. According to Webb et al. [122], involving a diverse group of stakeholders in multi-level governance will facilitate collaboration among institutions and jurisdictions, which is critical for addressing sustainability and resilience in complex urban systems. AG can respond to the uncertainty and complexity of complex socio-ecological systems and increase the capacity of UGI to tackle social and environmental circumstances such as the COVID-19 pandemic.

This approach mainly focuses on the concepts of flexibility, connectivity, and learning in form of policies, formal mechanisms, and regulatory standards which are required to maintain small-scale UGI such as stormwater management systems or rain gardens. Flexibility in adaptive governance arrangement allows stakeholders to adapt their needs and expectations to new opportunities and drive UGI regulations and policies [115]. This approach presents many opportunities for leveraging local UGI to support ES. For instance, converting vacant land into stormwater management systems requires flexible and responsible management structure tools, such as land banks to help municipalities identify suitable vacant parcels [123]. Without the flexibility of AG, it would be time-consuming and difficult for municipal agencies to respond to social and environmental disturbances [27].

Close connections among different actors facilitate negotiations and collaborations across horizontal (collaborative) and vertical (hierarchical) connections. In all types of small-scale UGI bottom-up and top-down connections are needed because city governments may support the establishment of community gardens, vacant lands, trees, rain gardens, stormwater management sites, and green roofs through the provision of land, funding, and regulations, whereas local communities may engage, utilize, and maintain them. Multiple actors are linked through various mechanisms such as collectively managed urban green spaces [38,124].

Social learning is part of the collaborative process of small-scale UGI adaptive governance in which actors can share their values, experiences, actions, and socio-ecological memories of ES through an active process of both formal and informal reflection [125–127]. For example, Lin and Egerer [128] emphasized the role of social learning as one of the characteristics of AG that supports food provision services in community gardens in which farmers learn through their experiences (e.g., light and water availability, soil properties, and ground cover management). Moreover, AG supports a cycle of monitoring and learning to better support ES and reduce disservices.

### 4.4. Mosaic Governance

The concept of mosaic governance (MG) in the context of UGI elevates the role of a diverse range of active citizens in UGI planning and citizen-led greening management and initiatives [129,130]. In this review, MG was mentioned in about 30 documents out of 91 analyzed. According to Buijs [131], active citizenship is defined as "*citizens' ability to organize themselves in a multiform manner, to mobilize resources and to act in the public to protect rights and take care of common goods*" (p.1). It may be separate from or connected to local authorities' arrangements [132]; however, local authorities usually can support active citizen practices in UGI. Active citizenship can leverage local UGI in the face of a financial crisis. For instance, during the recent pandemic and its consequent social and economic restrictions, active local citizens in community gardens might play a significant and creative role in managing, producing, and marketing local food for the neighborhoods [133]. Mattijssen et al. [134] propose a new term "green self-governance" to respond to the critical points including equal representation of non-active citizens and instrumentalization of citizens in the active citizenship approach. Green self-governance is defined as "a specific form of governance in which citizens play a major role in realizing, protecting, and/or managing green public space" [134]. In contrast to traditional centralized governance approaches for UGI, self-governance arrangements aim to strengthen the autonomy of citizens to shape their bottom-up initiatives and rules (e.g., citizen maintenance of local vacant lands).

MG approaches emphasize the connectivity and multifunctional nature of UGI and use reflexive notions of stewardship to facilitate the active citizenship process. Connectivity in mosaic-oriented governing of green infrastructure's ES involves two dimensions. First, linking UGI can create support and protect the functions and benefits of mosaic UGI that individual UGI cannot provide alone. Second, the connectivity between local authorities and residents is critical to the success of the UGI regime to foster social and ecological resilience. Gulsurd et al. [135] called this approach "reflexive co-governance", in which

citizens play an active role in creating healthy ecosystems. In contrast to traditional governance approaches, in MG or reflexive arrangements, both local government and local citizens can contribute to the delivery and management of ES in small-scale UGI. In this approach, local authorities recognize the legitimacy and autonomy of individuals as active citizens [131]. Thus, this approach includes polycentric governance in which multiple centers of governance interact with each other across diverse scales and actors. It also emphasizes environmental stewardship at local levels, where a self-organized approach to responding to small-scale heterogeneity (e.g., size, quality, range of activities, etc.) is highly recommended within a collaborative management activity [136–142]. According to some studies [143–145], contemporary urban environmental stewardship activity allows residents to protect and manage small parks, trees, and community gardens that provide ES. For example, Langemeyer et al. [146] found that civic urban gardens as a new way of connected UGI can enhance stewardship action much better rather than traditional allotment gardens.

### 4.5. Networked Governance

Networks can facilitate the relationship between formal governance arrangements and the informal social processes that affect the local governance of UGI. Of the 10 papers where it was addressed, only 5 of them defined networked governance (NG). According to Nochta and Skelcher [147], networked governance is defined as "*systems of coordination that seek to guide and steer multi-actor interactions to solve complex public policy problems*". Network governance of small-scale UGI creates a space for cooperation between various actors to facilitate the co-production of UGI policies based on knowledge sharing, which in turn can increase social and ecological resilience [148]. A community-based management approach is one example of networked governance that includes collective action and self-governance of ES that relies on strong connections between stakeholders at local levels [149]. For example, the networks among community gardeners help them increase social resilience and capital through sharing knowledge about food productivity essentials such as soil contaminant risks. Although networks may create unevenness in the distribution of power and resources [150], mismatches with ecological scales [139], and formal or informal dialogues [151,152], they are powerful tools to build and maintain community garden development [153].

Some scholars propose the networked governance approach as a type of adaptive governance in which the characteristics of actors and the patterns of interactions between different actors' matter and are important for improving the performance of ecosystems [154,155]. Key to this approach is the concept of diversity, which means the presence of various actors, often multi-level, are involved in the process of local green infrastructure planning [156]. For instance, a networked governance approach to street tree planting and maintenance requires the involvement of different actors including local municipalities, non-profit organizations, community groups, and individual volunteers. Some of these actors (e.g., municipalities and NGOs) mainly play financial and supportive roles, whereas others (e.g., individuals and community groups) may participate in tree planting and stewardship activities. The majority of UGI governance networks, such as residential green roofs, are informal and differ from traditional hierarchical modes of government [152]. Additionally, the concept of "ecological networks" is central to understanding the linkages between network governance, ES, and UGI. Ecological networks are defined as "tools for improving biodiversity and ecological connectivity among habitats which are designed to consider different levels of nature protection" [157]. This definition is also aligned with the concept of green infrastructure as a network of green spaces that provide multiple benefits.

More recently, a concept of hybrid governance has also been applied to UGI specifically nature-based solutions (NBS) governance. Toxopeus et al. [158] define hybrid governance as a tool "to drive innovation and deliver co-benefits to multiple stakeholders, representing a demand-driven, cost-effective realization of sustainable urban infrastructure". Hybrid governance as a type of polycentric and multi-level governance runs parallel to the net-

worked governance approach in local UGI due to its participatory and democratic nature. For example, over the last decade, public and non-public actors (e.g., businesses) in the U.S. context have collaborated to create the financial resource and technical synergies to install green roofs and solar PV. However, non-public actors have not had a role in the development of policies and regulation processes, which in turn has raised the issue of equity [159].

*4.6. Transformative Governance*

The literature regarding the application of transformative governance (TG) in the context of UGI is limited. Only three articles directly focused on the application of TG in the context of UGI. Transformative governance is defined as "*an approach to environmental governance that has the capacity to respond to, manage, and trigger regime shifts in coupled social-ecological systems (SESs) at multiple scales*". [25]. While the adaptive governance approach emphasizes maintaining and adapting existing ecosystem regimes, transformative governance arrangements pursue regime shifts to create new systems to better support ecosystems and human health and well-being. Regime shifts, as direct consequences of cross-scale dynamics of socio-ecological systems, can result in drastic transformative changes in different aspects of ES that UGI can provide at a local level called "panarchy", which is central to the concept of transformation [25]. Panarchy theory stresses cross-scale links, in which processes at one scale influence operations at other scales, influencing the system's overall dynamics [160]. For example, the recent COVID-19 outbreak occurred quickly and became widespread, which subsequently created considerable changes in food security, nutrition, and food systems worldwide. As Mejia et al. [133] state, the community gardens have had a significant role during the pandemic by providing fresh food and supporting human well-being and social benefits. According to this concept, small-scale UGI is considered in some cases (e.g., rain gardens) as a transformative governance response to climate change issues at a large scale and as an example of an adaptive governance approach in other situations (e.g., street trees).

Given the complex nature of small-scale UGI, TG aims to alter the nature of UGI through innovative approaches, such as nature-based solutions (NBS). Frantzeskaki and Bush [161] indicate the role of intermediaries (systemic, regime-based, niche, process, and user) as facilitators of transformative governance in the case of urban trees in Australia. This approach leverages systematic changes in UGI systems to address social and ecological challenges such as climate change. In small-scale UGI, examples of transformative changes may include the greening of vacant lands or the shifts from gray to green roofs. For instance, Kabisch [162] explains how local participation as a key feature of new governance modes can transform an urban brownfield site into a multifunctional urban park in Leipzig, Germany.

Transformative governance is considered part of transition management. Transition management has been defined as "a specific form of multi-level governance. Whereby state and non-state actors are brought together to co-produce and coordinate policies iteratively and evolutionarily on different policy levels" [163]. Loorbach [164] describes the tenets of transition management as (1) long-term focus, (2) uncertainties and surprise, (3) networks and self-steering, and (4) innovation. Social innovation is a new term used for environmental governance and management in the transition arena. It refers to a set of new concepts, products, and processes that not only seek to meet basic social needs but also change routine flows and arrangements to better utilize the urban and social systems [165]. Green infrastructure can be considered an innovative response to complex and challenging urban systems and can support transformative governance [166].

Social innovation is a process of change that can facilitate the emergence of transformative governance in the context of UGI. According to Elmqvist et al. [167], social innovation can play an important role in urban transformations by proposing new alternatives and configurations for preserving the functionality of a system such as UGI. For example, community gardens serve as an example of social innovation because they

enhance the value of provisioning and regulating ES such as food production and air quality improvements [142]. Social innovation at the local scale can create physical and community space for optimizing the usage of UGI for ES. For instance, transforming vacant lands into UGI can provide opportunities for provisioning, regulating, and socio-cultural ES [72,102]. Likewise, due to the unused nature of urban vacant lots and the greening opportunities, they are mostly documented under the transformative governance approach in the literature. Nevertheless, there is room to examine their applicability for implementing some temporary socio-cultural services, short-term activities or policies (e.g., land bank in the U.S.) which can be matched with the adaptive governance approach [27,102,123]. More recently, digital tools such as smart ecosystems and the internet of trees as digital networks are new concepts that can support the environmental transformative governance approach [168,169].

Our analysis showed that all four networks, mosaic, transformative, and adaptive governance approaches are applicable for different provisioning, regulating, cultural, and supporting ES through practicing principles in six small-scale UGI, as shown in Table 2; however, these occur to varying degrees, depending on their principles' applicability in UGI cases. In other words, each governance mode can be effective for managing the ES of certain small-scale UGI, given their relationships with principles. First, the principles of resilience thinking and connectivity are interconnected with food and water supply in UGI governance. For example, social resilience can be developed using connectivity concepts between different stakeholders and governments to produce and distribute food through community gardens. Connectivity among diverse actors can support mosaic and adaptive governance arrangements as well, which can build the resilience of ES through social networks, preventing disturbances, and maintenance of biodiversity [165]. Second, involving citizens and integrating citizen science concepts in small-scale UGI as part of mosaic governance can address cultural ES in some cases, such as developing a small new pocket park with volunteers [73]. The literature shows that active citizenship has a significant role in governing cultural ES such as environmental education, recreation, or social cohesion, in turn, can promote mental and physical health outcomes [132]. Third, the concept of socio-ecological networks, which are central to governance networks, can enhance both vertical and horizontal connectivity between multi-level government arrangements and habitats, respectively, to support biodiversity conservation practices. For example, actors (e.g., government, citizens, or NGOs) can balance supply and demand in green spaces by conserving native plants or using interventions such as "community-based management" which can create new supply nodes connected to demanders [149].

Our study has a few limitations. This review did not directly analyze the political and financial leadership and arrangement concepts in the management of governance of small-scale UGI. It is possible that the examples mentioned from different geographical areas represent the type of political system or governance and may not reflect the conditions of other existing examples well. Therefore, further analysis in terms of clarifying the potential interlinkages between government and political structures, and their capacities for governing local UGI is needed [170]. Furthermore, we did not identify the potential differences between the four modes of governance in terms of how they might address and resolve issues related to EDS. This approach needs to be further analyzed by highlighting synergies and tradeoffs between each UGI type, the applicable governance approach, and examining the crossovers and combined use of the four governance approaches in different ways [116]. It should be noted too that we could not provide more empirical evidence of analysis of the situation about small-scale UGI due to the word number limitations for publication. Finally, our review did not identify the best governance mode for each type of ES/EDS associated with UGI, as well as the characteristics of a successful governance structure, which can be investigated based on more empirical studies.

Figure 4 shows a Sankey diagram mapping the systematic review of the literature regarding the potential linkages between new governance modes and six types of UGI. As is shown, vacant lots and green roofs were among those small-scale UGI which were

less documented and there is a need to address this gap by applying new governance arrangements to optimize their benefits for urban environments and residents. To advance this approach, more studies are needed to evaluate potential links between different governance modes and their associated ES in different contexts and types of UGI. Additionally, future studies can focus on the applicability and suitability of governance approaches to offer equity, resilience, and sustainability concepts in the face of climate change and societal disruptions such as the COVID-19 pandemic.

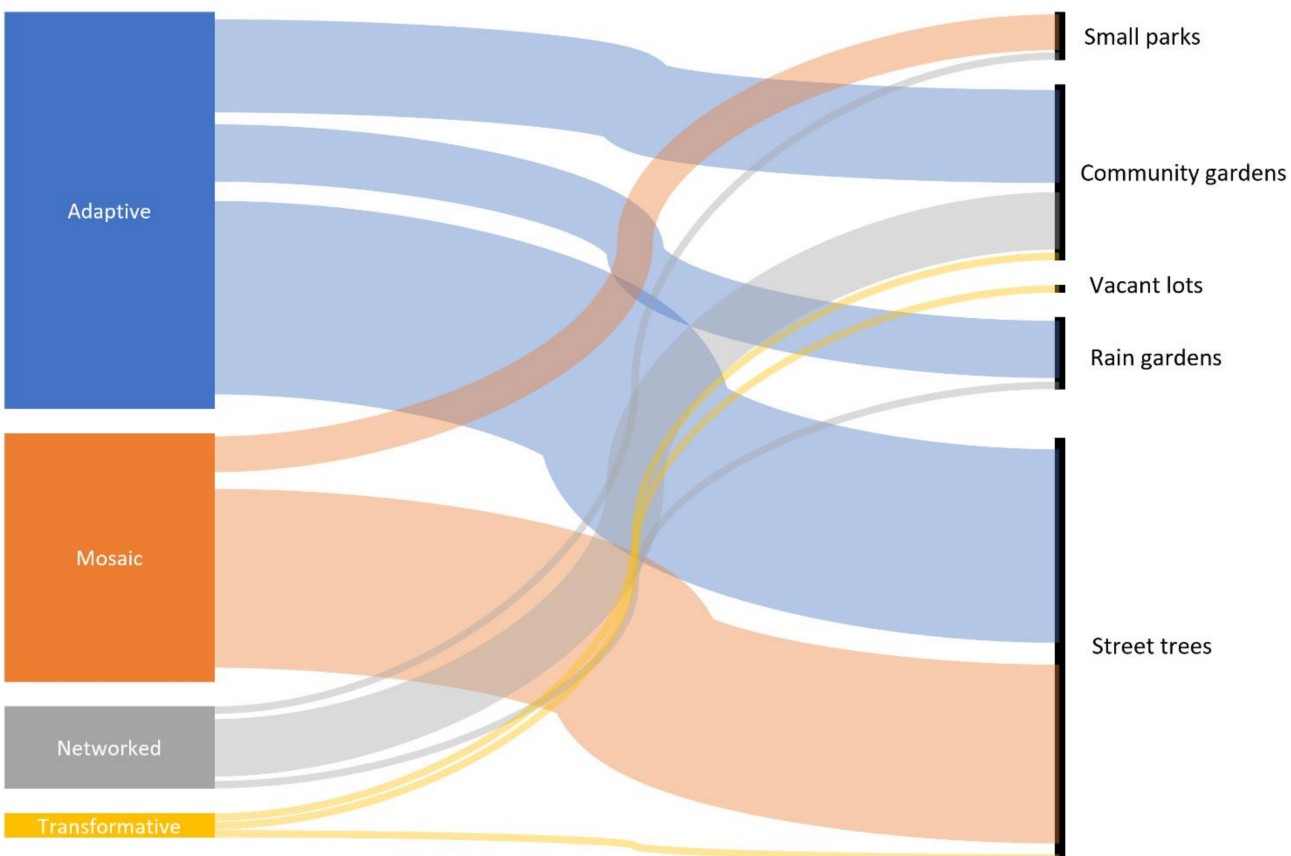

**Figure 4.** Sankey diagram showing the link of governance modes and UGI types. The thickness of each band corresponds to the number of studies involving the linkage.

## 5. Conclusions

Our review provides a first overview of the applicable governance approaches for six types of small-scale green infrastructure including small parks, community gardens, vacant lots, rain gardens, green roofs, and street trees. This review focused on the most common governance approaches, including adaptive, mosaic, networked, and transformative approaches for managing the ES and EDS of selected small-scale UGIs. This study provided some insights for policymakers, planners, researchers, and those who are interested in investigating the linkages between UGI governance approaches and ES. First, it demonstrated a novel attempt to categorize new governance approaches for managing ES in the context of small-scale UGI. This review showed that each governance approach had its and specific principles for framing ES in different small-scale UGI which are moderated by contextual factors conducting nuanced linkages between governance approaches and ESES. Second, as our review indicated that the specific characteristics of small-scale UGI such as location, ownership, the type of provided ES, and their potential for further development may determine the mode of governance. However, cultural, political, economic, and government structures in different contexts may affect this relationship to some degree. It is extremely important to examine different governance approaches in diverse contexts, as well as the potential of combined governance modes for other UGI types. In other words,

the flexible adoption of different governance arrangements rather than only selecting a certain mode may become useful for managing multifunctional UGI types or adapting to environmental and societal change.

Finally, although there are multiple studies regarding different governance modes in natural resources management at national or state scales, this study highlighted the current status quo of knowledge and future potential research directions regarding the linkages between diverse new governance types and ES/EDS of small-scale UGI. The research gaps can help urban planners, green infrastructure planners, and urban ecologists pursue suitable governance approaches for better utilizing different types of services provided by small-scale GI in ES within cities.

**Author Contributions:** Conceptualization, methodology, and writing were carried out by S.R.A.; and H.P. reviewed and edited previous draft versions and provided supervision. All authors have read and agreed to the published version of the manuscript.

**Funding:** This research received no external funding.

**Institutional Review Board Statement:** Not applicable.

**Informed Consent Statement:** Not applicable.

**Data Availability Statement:** Data are available from the first author on request.

**Acknowledgments:** The authors acknowledge and thank the helpful comments provided by an anonymous reviewer.

**Conflicts of Interest:** The authors declare no conflict of interest in the manuscript, or in the decision to publish the results.

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
