# Peer review of "How Do Different Modes of Governance Support Ecosystem Services/Disservices in Small-Scale Urban Green Infrastructure? A Systematic Review"

_land, doi:10.3390/land11081247_

Round 1

Reviewer 1 Report

Dear authors,

I do not have any comments to add because your paper is very clear.

Congratulations!

Author Response

Thanks so much for your review.

Reviewer 2 Report

How do different modes of governance support ecosystem services in small-scale urban green infrastructure?

General comments

This review is focused on a very timely topic with huge importance to the concept of the sustainable development of the cities to face some of the most important challenges in the next decades related to global change. Under my point of view, this study is very well written and structured and is based on a very well developed literature review on the topic of services of urban green spaces and governance approaches. From the methodological point of view, I think that the authors could adapt without effort the method of review of literature to the standardised PRISMA procedure for systematic reviews (referred below in specific comments).

The four governance types are well explained and extensively described in the last sections of the manuscript, but it would be recommendable if the main differences are described in the first lines to clarify these concepts to the reader before Table 2. As the authors recognise, this study does not discuss which is the best governance model for each urban area type, but it could be addressed in future researches in line with this study.

The most important negative point of the work under, my point of view, is that disservices of the urban green spaces are poorly addressed in this manuscript. The disservices are referred in several points, but these disservices are not well described. For instance, the authors have considered the disservices as a subsection of the ecosystem services (e.g. Table 1), which in my opinion, it should be extracted as another main section to be analysed. I encourage the authors to include several references on this regard, especially in pollen from allergenic ornamental plants as a very important problem of public health.

Addressing several aspects missed in the original manuscript, I think this work could be recommended for publication in this journal. The study has enough quality to be published after revisions.

Specific comments

- Line 6. Indicate the acronym between parenthesis: "urban green infrastructures (UGI)"

- Lines 13-15. Could you indicate better the four governance approaches. It is the first time that the reader finds the reference to the governance types that will be treated in the review.

- Lines 27-28. I do not think so that urbanization exacerbate climate change itself, perhaps better "exacerbate consequences of the climate change and reduce the adaptation capability to...". Please, rephrase.

- Lines 32. Please, talk about "urban heat island effect" and perhaps better "altered runoff patterns" instead of "altered precipitation patterns".

- Lines 33-38. The authors only describe ecosystem services of the urban green areas, although in line 40 they indicate "services and disservices". Doing a complete description of the relationship of green areas with human population requires also to consider the description of the disservices which are the subject of the study of publications such as Escobedo et al. 2011 (https://doi.org/10.1016/j.envpol.2011.01.010), which is relevant to be cited in this review.

Also, in my opinion, the most important disservice of urban green areas are those related to pollen release from ornamental wind-pollinated plants. This topic is important also to be cited (Maya-Manzano et al. 2017  https://doi.org/10.1016/j.ufug.2017.09.009)

- Line 58. "GI ranges...". Better "UGI ranges...". Maintain the same acronym along the entire text.

- Lines 58-60. Is this classification or urgan green spaces a standardised previous classification? Any reference?

- Lines 67-71. About the disservice of allergenic pollen release from ornamental flora, you may comment as very diverse small-scale green spaces could reduce allergenic risk from urban green spaces, with respect to large- and medium-scale green areas commonly with large surface of monospecific allergenic species in the cities (Cariñanos and Casares-Porcel 2011, https://doi.org/10.1016/j.landurbplan.2011.03.006).

- Lines 88-97. The review process of literature is fine, but I think that the current process would be easy to be adapted to the standardised PRISMA procedure to systematics reviews (Preferred Reporting Items for Systematic Reviews and Meta-Analyses) (Moher et al. 2009, https://doi.org/10.1371/journal.pmed.1000097).

- Line 118. Which scientific search engine has you used to carry out the literature review?

- Line 180, and line 190. Not only grass allergy is relevant from urban green spaces, in fact, arboreal plants used as ornamental species are more important in these managed spaces, as grasses are located in urban areas, peri-urban and rural areas. In the Mediterranean region, the ornamental patterns of the urban areas imply significant pollen risk from woody species such as plane trees or cypresses, as the most allergenic ornamental species (Lara et al. 2020,  https://doi.org/10.3390/f11080817; Pecero-Casimiro et al. 2020 https://doi.org/10.3390/rs12101562). And I disagree that "disservices have been less studied" when this aspect is a very timely topic and it has not been studied in detail in this manuscript (e.g. Sousa-Silva et al. 2021, https://doi.org/10.1038/s41598-021-89353-7).

- Line 193. Disservices in Table 1 are considered as one more subsection of services, and I think it should be extracted as a new section at the same level as "ecosystem services".

- Lines 201-202. The consequence of the change of land-use in vacant-lots may impact public health, in this case, in relation to allergenic weeds in spontaneous urban vegetation. Therefore, these risks should be considered in the management and governance of green urban areas. An interesting paper in this regard is Katz et al. 2014 (https://doi.org/10.1016/j.ufug.2014.06.001).

- Line 262. Could the authors introduce to the readers in the first paragraph the main characteristics of the four governance approaches that are analysed in this manuscript? It would be useful for the readers to discover the main differences that characterise the different approaches in the first lines.

Author Response

General comments

  • This review is focused on a very timely topic with huge importance to the concept of the sustainable development of the cities to face some of the most important challenges in the next decades related to global change. Under my point of view, this study is very well written and structured and is based on a very well-developed literature review on the topic of services of urban green spaces and governance approaches. From the methodological point of view, I think that the authors could adapt without effort the method of review of literature to the standardised PRISMA procedure for systematic reviews (referred below in specific comments).

R1: The PRISMA methodology was applied. We do not aim at doing meta-analyses in the case that exploring main concepts and research gaps is more of important than finding correlations between findings. In fact, this is an emerging topic that needs to be further analyzed through a comprehensive literature review like what we exactly did.  

  • The four governance types are well explained and extensively described in the last sections of the manuscript, but it would be recommendable if the main differences are described in the first lines to clarify these concepts to the reader before Table 2. As the authors recognise, this study does not discuss which is the best governance model for each urban area type, but it could be addressed in future research in line with this study.

R2: Some general statements regarding the overlaps and differences of four modes were included in the result section, although sometimes authors have used them interchangeably in literature which needs to be more investigated in future research. The objective of this study is not to compare these modes and find their differences or similarities, rather we aimed at exploring their characteristics and principles used in the case of small UGI.

  • The most important negative point of the work under, my point of view, is that disservices of the urban green spaces are poorly addressed in this manuscript. The disservices are referred in several points, but these disservices are not well described. For instance, the authors have considered the disservices as a subsection of the ecosystem services (e.g. Table 1), which in my opinion, it should be extracted as another main section to be analysed. I encourage the authors to include several references on this regard, especially in pollen from allergenic ornamental plants as a very important problem of public health.

 R3: To address the reviewer’s concern regarding ecosystem disservices, we add a new section to cover this topic.

 Specific comments

 - Line 6. Indicate the acronym between parenthesis: "urban green infrastructures (UGI)"

R4: It was done in line 7.

- Lines 13-15. Could you indicate better the four governance approaches. It is the first time that the reader finds the reference to the governance types that will be treated in the review.

R5: It was done in lines 15-19.

 - Lines 27-28. I do not think so that urbanization exacerbate climate change itself, perhaps better "exacerbate consequences of the climate change and reduce the adaptation capability to...". Please, rephrase.

 R6: It was done in lines 33-34.

- Lines 32. Please, talk about "urban heat island effect" and perhaps better "altered runoff patterns" instead of "altered precipitation patterns".

 R7: It was done in line 39.

- Lines 33-38. The authors only describe ecosystem services of the urban green areas, although in line 40 they indicate "services and disservices". Doing a complete description of the relationship of green areas with human population requires also to consider the description of the disservices which are the subject of the study of publications such as Escobedo et al. 2011 (https://doi.org/10.1016/j.envpol.2011.01.010), which is relevant to be cited in this review.

Also, in my opinion, the most important disservice of urban green areas are those related to pollen release from ornamental wind-pollinated plants. This topic is important also to be cited (Maya-Manzano et al. 2017  https://doi.org/10.1016/j.ufug.2017.09.009)

 R8: New citations were added.

- Line 58. "GI ranges...". Better "UGI ranges...". Maintain the same acronym along the entire text.

 R9: It was done.

- Lines 58-60. Is this classification or urban green spaces a standardised previous classification? Any reference?

R10: Our goal to state this sentence was stressing on the multiscalar nature of UGI. Thus, according to an article recently reviewed this feature of UGI, we add a new citation in line 72.

 - Lines 67-71. About the disservice of allergenic pollen release from ornamental flora, you may comment as very diverse small-scale green spaces could reduce allergenic risk from urban green spaces, with respect to large- and medium-scale green areas commonly with large surface of monospecific allergenic species in the cities (Cariñanos and Casares-Porcel 2011, https://doi.org/10.1016/j.landurbplan.2011.03.006).

 R11: The introduced citation as well as your note was added in lines 79-83.

 - Lines 88-97. The review process of literature is fine, but I think that the current process would be easy to be adapted to the standardised PRISMA procedure to systematics reviews (Preferred Reporting Items for Systematic Reviews and Meta-Analyses) (Moher et al. 2009, https://doi.org/10.1371/journal.pmed.1000097).

  R12: We moved the methods part to the before results section in order to better establish a general literature review to frame our discussion. Also, as I mentioned before, we added the PRISMA method to the methods part. New descriptive analysis in the result section including date, type of studies, types of UGI, and types of governance modes were added. The research methods part was categorized in some subcategories.

- Line 118. Which scientific search engine has you used to carry out the literature review?

 R13: As it is mentioned in line 372, Web of Science is the main search engine.   

 - Line 180, and line 190. Not only grass allergy is relevant from urban green spaces, in fact, arboreal plants used as ornamental species are more important in these managed spaces, as grasses are located in urban areas, peri-urban and rural areas. In the Mediterranean region, the ornamental patterns of the urban areas imply significant pollen risk from woody species such as plane trees or cypresses, as the most allergenic ornamental species (Lara et al. 2020,  https://doi.org/10.3390/f11080817; Pecero-Casimiro et al. 2020 https://doi.org/10.3390/rs12101562). And I disagree that "disservices have been less studied" when this aspect is a very timely topic and it has not been studied in detail in this manuscript (e.g. Sousa-Silva et al. 2021, https://doi.org/10.1038/s41598-021-89353-7).

 R14: the new citations were added in lines 238-240.

 - Line 193. Disservices in Table 1 are considered as one more subsection of services, and I think it should be extracted as a new section at the same level as "ecosystem services".

 R15: A new table 2 was added.

 - Lines 201-202. The consequence of the change of land-use in vacant-lots may impact public health, in this case, in relation to allergenic weeds in spontaneous urban vegetation. Therefore, these risks should be considered in the management and governance of green urban areas. An interesting paper in this regard is Katz et al. 2014 (https://doi.org/10.1016/j.ufug.2014.06.001).

  R16: The citation was added in lines 254-257.

- Line 262. Could the authors introduce to the readers in the first paragraph the main characteristics of the four governance approaches that are analysed in this manuscript? It would be useful for the readers to discover the main differences that characterise the different approaches in the first lines.

  R9: It was explained in lines 417-422.

Round 2

Reviewer 2 Report

In my opinion, the manuscript has improved and the authors addressed point-by-point all my previous comments. My main objection related to disservices has been completely addressed in this version in a perfect way.

Congratulations